# The Nocardia Rubra Cell Wall Skeleton Regulates Macrophages and Promotes Wound Healing

Kai Hu [1,†], Yan Xu [2,†], Xiaoxiao Li [1], Pan Du [3], Yichi Lu [3] and Guozhong Lyu [1,2,*]

1 Nanjing University of Chinese Medicine, Nanjing 210023, China
2 Affiliated Hospital of Jiangnan University, Wuxi 214041, China
3 Jiangnan University, Wuxi 214122, China
* Correspondence: luguozhong@hotmail.com or luguozhong@jiangnan.edu.cn
† These authors contributed equally to this work.

**Abstract:** The Nocardia rubra cell wall skeleton (Nr-CWS) is an immunomodulator used clinically for its ability to modulate the body's immune function. Macrophages are an important hub of the immune response during wound healing. In this study, we hypothesized that a Nr-CWS could modulate macrophage physiological activities, polarize macrophages toward M2, and promote wound healing. Through in vivo experiments, we made two full-thickness excisional wounds on the backs of mice; one was treated with a Nr-CWS, and the other was treated with saline. We photographed and recorded the wound change every other day. We observed the histopathological examination and collagen deposition using H&E and Masson staining, then analyzed the macrophage surface markers using immunofluorescence. Through in vitro experiments, we studied the effect of the Nr-CWS on RAW264.7 cells through CCK8, transwell, flow cytometry, western blot, immunofluorescence, and ELISA. We found that the Nr-CWS can enhance the proliferation, migration, and phagocytosis of macrophages. In addition, it can promote the recruitment of macrophages on the wound surface, polarize macrophages to M2, and increase the expression of pro-healing cytokines. Ultimately, the Nr-CWS accelerated wound healing.

**Keywords:** macrophage; polarization; Nr-CWS; inflammation; wound

## 1. Introduction

The Nocardia rubra cell wall skeleton (Nr-CWS) was extracted from a rubra, a gram-positive bacterium that consists of nocardia acid, arabinogalactan, and mucopolysaccharide. Previous studies have shown that Nr-CWSs can act as an immunomodulator to upregulate the body's immune function for antitumor purposes, and it can also accelerate the wound repair process in patients with celiac disease [1]. Some researchers have applied it to mouse skin wounds and found that it can promote healing [2].

The immune system plays a crucial role in the wound healing process, and the highly complex immune network includes the synergistic action of multiple immune cells. The immune cells can produce various cytokines to affect skin wound healing [3]. Wound macrophages and the associated cytokines they secrete are at the center of this immune network. Under the chemotactic effect of inflammatory factors, blood monocytes cross the vascular endothelial cell layer into the peritraumatic tissue. They are activated into macrophages through a series of reactions. Macrophages can be classified into two different phenotypes, M1 and M2, depending on their activation mode and function. M1 macrophages are classically activated and usually accumulate at the wound surface during the early inflammatory phase of wound healing, and M2 macrophages are alternatively activated and typically gather at the wound site during the repair phase of wound healing. This transformation is necessary for wound healing [4]. The inhibition of macrophage recruitment to the wound site or phenotypic deficiencies can affect the quality and speed of

wound healing [5]. In this work, we explored the effect of a Nr-CWS on wound immune response, primarily in macrophages.

## 2. Materials and Methods

### 2.1. Chemicals and Reagents

The Nr-CWS was provided by Liaoning Greatest Bio-Pharmaceutical Co., Ltd. The high glucose medium, fetal bovine serum, and 0.25% EDTA trypsin were purchased from Gibco (San Jose, CA, USA). The streptomycin-penicillin was purchased from Thermo (Waltham, MA, USA). The PBS, CCK-8 kit and RNA rapid extraction kit were purchased from Yishan (Shanghai, China). The 0.8 μM transwell chambers, DMSO, DAPI, Alexa Fluor 488 Goat anti-mouse, β-actin antibody, and GAPDH antibody were purchased from Sigma (St. Louis, MO, USA). The FITC-Dextran (MW4000) was purchased from MCE (Princeton, NJ, USA). The TNF-α and TGF-β ELISA kits were purchased from Lianke Bio (Hangzhou, China). The Arg-1 and iNOS antibodies were purchased from Bio Legend (San Diego, CA, USA). The H&E staining kit, quantitative PCR kit, and reverse transcription kit were purchased from Yeason (Shanghai, China). The Masson staining Kit was purchased from Sbjbio (Nanjing, China). The 4% paraformaldehyde, anti-fluorescence quenching blocking solution, and immunohistochemical blocking solution were purchased from Beyotime (Shanghai, China). The primers were purchased from Sangon Biotech (Shanghai, China). The mTOR/p-Mtor, Akt/p-Akt, PI3K/p-PI3K, and TGF-β antibodies were purchased from Abcam (Cambridge, UK).

### 2.2. In Vivo Experiments

2.2.1. Experimental Healing of Full-Thickness Excisional Wounds in Mice

Six- to eight-week-old C57BL/6J female mice were purchased from the SLAC laboratory in Shanghai and housed at the Jiangnan University Medical Experimental Animal Center. The Animal Ethics Committee of Jiangnan University approved all of our experimental animal procedures. Animal Ethics Approval Number: JN.No20210415c1350920[086]. A mouse model of full-thickness excisional wounds was prepared according to a previously described method [6]. Thirty female C57BL/6J mice, weighing 20–25 g, were used in the experiment. After seven days of feeding, we removed the hair from their backs with hair removal cream, the mice were anesthetized with 40 mg/kg sodium pentobarbital by intraperitoneal injection, and two circular-shaped full-thickness excisional wounds with a diameter of 6 mm were cut out of their backs. The mice were randomly divided into two groups. The Nr-CWS (60 micrograms) was dissolved in 2 mL saline. All subsequent experiments were the same. Then, we injected the saline and Nr-CWS uniformly into four sites of the wound bed, 20 μL at a time, and then covered it with 3 M film (1624 W, 3 M), protected with a layer of medical tape to prevent drying and infection. We changed the fresh dressing for the wounds and photographed them every other day, as well as calculating the wound healing rate and wound area by ImageJ.

2.2.2. Histopathological Examination

Wound specimens were taken on Days 3, 7, and 10 after surgery, fixed in 4% formaldehyde and PBS buffer (pH 7.2) and embedded in paraffin. The 5 μm thick tissue sections were stained for H&E and modified Masson trichrome according to the kits' instructions [7].

2.2.3. Immunofluorescence

On Days 3, 7, and 10, the tissue sections were dewaxed and hydrated. The sections were then immersed in sodium citrate antigen repair solution at 100 °C for 25 min for antigen repair. We blocked the sections with immunohistochemical blocking solution for 60 min at room temperature, and then CD163 and CD68 primary antibodies were applied, according to the instructions, and incubated overnight at 4 °C. The next day, we set the sections in the secondary antibody for 90 min at room temperature with three washes of

PBST, for 5 min each. We took images by laser confocal microscopy and analyzed them using ImageJ.

### 2.2.4. Wound Healing Rate

Wound images were taken with a digital camera on Days 0, 2, 4, 7, and 10 for general observation and wound closure analysis. The formula for wound healing rate was calculated as follows: (original wound area − current measured area) ÷ original wound area × 100%.

### 2.3. *In Vitro Experiments*

### 2.3.1. Cell Culture

The primary cells (RAW 264.7, mouse-derived macrophage cell line) were purchased from Wuhan Procell. Cells of the 3rd–5th generation were maintained in high glucose medium containing 10% fetal bovine serum and 1% penicillin-streptomycin at 37 °C with 5% $CO_2$ and changed the culture medium every 24 h [8].

### 2.3.2. Cell Viability Assay

Cell activity was analyzed using a CCK-8 kit. The RAW264.7 cells ($1 \times 10^4$ cells/well) were plated on 96-well plates and placed in the incubator for 24 h. The Nr-CWS (30 μg/mL) was added to the well plates at 0%, 5%, 10%, 25%, and 50% concentrations and incubated for 24 h, protected from light, and placed in the incubator for 1 h after adding CCK-8 reagent. Finally, we detected the absorbance at 450 nm on the enzyme maker. (ELX800 Bio-Tek, Venusky, VT, USA).

### 2.3.3. Cell Phagocytosis

The Raw264.7 cells ($1 \times 10^4$ cells/well) were plated on 96-well plates, incubated with DMEM and 5% Nr-CWS in DMEM for 24 h. Then, the FITC-Dextran was added to protect the cells from light for 1h. We stopped the reaction with cold phosphate buffer (PBS) containing 2% fetal bovine serum and washed three times; resuspended in PBS containing 4% paraformaldehyde. The impact of the Nr-CWS on the phagocytosis of the dextran particles in the macrophages was analyzed by flow cytometry. (Accuri C6 Plus BD, San Jose, CA, USA) [9].

### 2.3.4. Cell Migration

For the transwell migration assay, the RAW264.7 cells ($3 \times 10^4$ cells/well) were plated in the upper chamber of a 0.8 μM diameter transwell 24-well plate, DMEM and DMEM containing 5% Nr-CWS were added to the lower chamber. After 12 h of incubation, the upper chamber was removed, and the cells on the surface of the upper chamber were wiped off. The cells on their outer surface were stained with 0.1% crystalline violet for 15 min, washed three times with PBS, then observed under an ordinary light microscope and photographed.

### 2.3.5. Cell Immunofluorescence

The cells were incubated for 24 h in a 12-well plate with cell supports, divided into a DMEM group and Nr-CWS group, and incubated for 24 h. The cells were washed three times with PBST, permeabilized with 0.5% Triton X-100 (Sigma, St. Louis, MO, USA) for 1 h, and then blocked at room temperature for 1 h. We incubated the cells overnight with a primary antibody; then, they were incubated with a secondary antibody at room temperature for 2 h. The cell nuclei were stained with DAPI. Photos were taken using an inverted Leica phase contrast microscope. Three photographs were taken from a random field. We analyzed the average fluorescence intensity by ImageJ.

### 2.3.6. Flow Cytometry

To detect the intracellular markers Arg-1 and iNOS, the cells were incubated with antibodies Arg-1 and iNOS on ice for 45 min and then washed and analyzed by flow cytometry (Accuri C6 Plus BD, San Jose, Ca, USA) [10].

### 2.3.7. RNA Extraction and Quantitative Real-Time PCR (qPCR)

The total RNA was extracted from two groups of RAW264.7 cells using the RNA Rapid Extraction Kit. The concentration of RNA was measured using a NanoDrop system from Thermo Fisher Scientific, Waltham, MA, USA. Then, the cDNA was amplified from 2 μg of total RNA using a fluorescent quantitative PCR kit, and a real-time quantitative polymerase chain reaction was performed on a Light Cycler 480II (Roche, Rheinland) using SYBR green. GAPDH was used as an internal reference. Primer sequences for other genes are shown in Table 1. The ratio of the relative expression of each gene to the GAPDH value was calculated using the $2^{-\Delta\Delta CT}$ formula [11].

**Table 1.** Primers used for quantitative reverse transcription-polymerase chain.

| Gene | Sequences (5′-3′) | Product Size (bp) |
|------|-------------------|-------------------|
| iNOS | forward:ATCTTGGAGCGAGTTGTGGATTGTC | 146 |
| | reverse:TAGGTGAGGGCTTGGCTGAGTG | |
| Arg-1 | forward:AGACAGCAGAGGAGGTGAAGAGTAC | 118 |
| | reverse:AAGGTAGTCAGTCCCTGGCTTATGG | |
| IL-10 | forward:AGAGAAGCATGGCCCAGAAATCAAG | 136 |
| | reverse:CTTCACCTGCTCCACTGCCTTG | |
| TGF-β1 | forward:ACCGCAACAACGCCATCTATGAG | 91 |
| | reverse:GGCACTGCTTCCCGAATGTCTG | |
| GAPDH | forward:TGACATCAAGAAGGTGGTGAAGCAG | 224 |
| | reverse:GTGTCGCTGTTGAAGTCAGAGGAG | |

### 2.3.8. ELISA

The P4 generation RAW264.7 cells were used and divided into DMEM and Nr-CWS groups. After a 24 h incubation, we discarded the old medium and washed the cells three times using PBS. DMEM was added and incubated for 12 h, then the supernatant of RAW264.7 was collected from each group, centrifuged at $350\times g$ for 5 min to remove cell debris, and stored at −80 °C. ELISA was performed according to the kit instructions.

### 2.3.9. Western Blot Analysis

The Petri dish was placed on ice, the medium was aspirated, and the cells were washed three times with PBS. The ready-made protein lysis solution (RIPA: PMSF: phosphatase inhibitor: protease inhibitor = 1000:10:10:1) was added, and the cells lysed on ice for 5 min. A cell scraper was used to scrape up the protein from the bottom of the dish, and it was then collected into a 1.5 mL EP tube. After ultrasonic lysis and centrifugation at 4 °C for 15 min at 12,000 r·min$^{-1}$, the supernatant was carefully aspirated, and the sample protein was prepared according to the volume ratio of supernatant: 5 × SDS loading buffer = 4:1. The protein was denatured at 100 °C for 20 min, and then brought to room temperature, naturally, before the sample protein was stored at −80 °C for subsequent experiments. The sample volume was calculated according to the sample volume of 20 μg for each protein group based on the measured protein concentration. Electrophoresis was performed at 80 V for 30 min and 120 V for 60 min. The proteins were then transferred to a membrane at 250 mA for 90 min, blocked at room temperature on a shaker for 60 min, and then incubated overnight at 4 °C on a shaker with a primary antibody. The membrane was washed three times with TBST for 5 min each time, incubated at room temperature on a shaker for 60 min

with a secondary antibody, and then washed three times with TBST, for 5 min each time. The strips were exposed and developed in the Bio-Rad gel imaging system, and the images were processed using ImageJ.

### 2.4. Statistical Analysis

All statistical analyses are expressed as the mean ± SEM using the GraphPad program ver. 6 (GraphPad Prism Software, Inc., San Diego, CA, USA). Statistical significance was determined using a paired *t*-test or a one(two)-way analysis of variance (ANOVA). $p < 0.05$ was considered to be significant. Each experiment was performed in triplicate and repeated at least three times.

## 3. Results

### 3.1. In Vitro Experiments

3.1.1. Nr-CWS Enhances the Proliferation, Migration, and Phagocytosis of RAW264.7 Macrophages

We first investigated the changes in the cell proliferation activity after applying different concentrations (0%, 5%, 10%, 25%, 50%) of the Nr-CWS on the RAW264.7 cells for 24 h. The proliferation activity of the RAW264.7 cells significantly increased with increasing concentrations (Figure 1A). To avoid the effect of proliferation, we then chose a 5% concentration of Nr-CWS for the subsequent experiments. Then, we measured the impact of Nr-CWS on the migratory activity of the RAW264.7 cells using a transwell assay. After 24 h, in comparison to the DMEM group, the number of migratory cells in the Nr-CWS group significantly increased (Figure 1B). Finally, we observed the effect of the Nr-CWS on RAW264.7 phagocytic activity, and the results were determined using a flow cytometer. The number of FITC-positive cells was significantly higher in the Nr-CWS group compared to the DMEM group (Figure 1C).

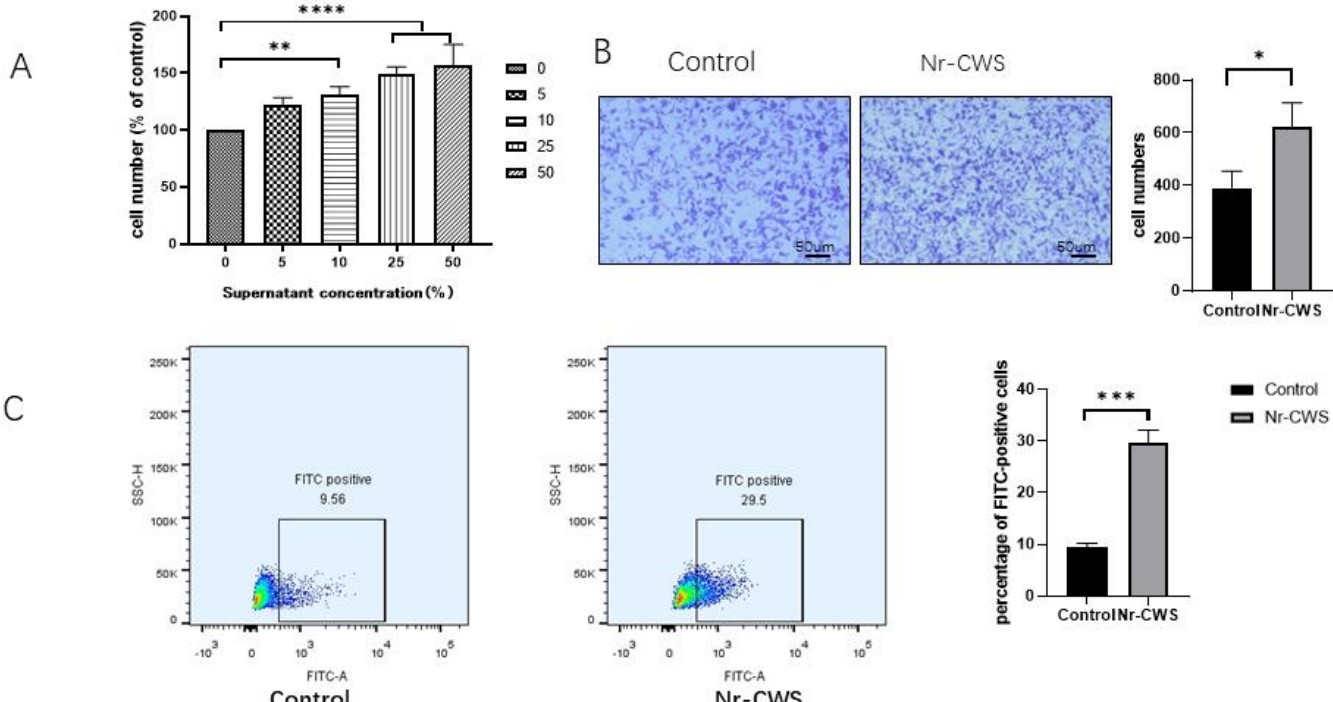

**Figure 1.** Nr-CWS regulates the proliferation, migration and phagocytic activities of RAW264.7 cells. (**A**) Effects of different concentrations of Nr-CWS on the proliferation activity of RAW264.7 cells. (**B**) Effect of 5% Nr-CWS on the migration activity of RAW264.7 cells. (**C**) Effect of 5% Nr-CWS on the phagocytosis of RAW264.7 cells by flow cytometry. (* $p < 0.05$, ** $p < 0.01$, *** $p < 0.001$, **** $p < 0.0001$ vs. control, paired *t* test and one-way ANOVA).

### 3.1.2. Nr-CWS Can Polarize Macrophages toward the M2 Type and Promote the Expression of Related Cytokines

We used flow cytometry and immunofluorescence to identify the macrophage phenotypes to examine the effect of the Nr-CWS on the polarization of the RAW264.7 cells. In this study, Arg-1 and iNOS were used as specific markers for the M2 and M1 phenotypes, respectively. The RAW264.7 cells were initially considered to be the macrophage Mø. Analysis by flow cytometry showed that the percentage of iNOS-positive cells decreased, while the percentage of Arg-1-positive cells increased upon Nr-CWS treatment (Figure 2A). The immunofluorescence results also showed that the number of Arg-1-positive cells increased dramatically while the number of iNOS-positive cells decreased after Nr-CWS treatment (Figure 2B). This finding suggests that Nr-CWSs can promote the polarization of RAW264.7 cells toward M2 macrophages.

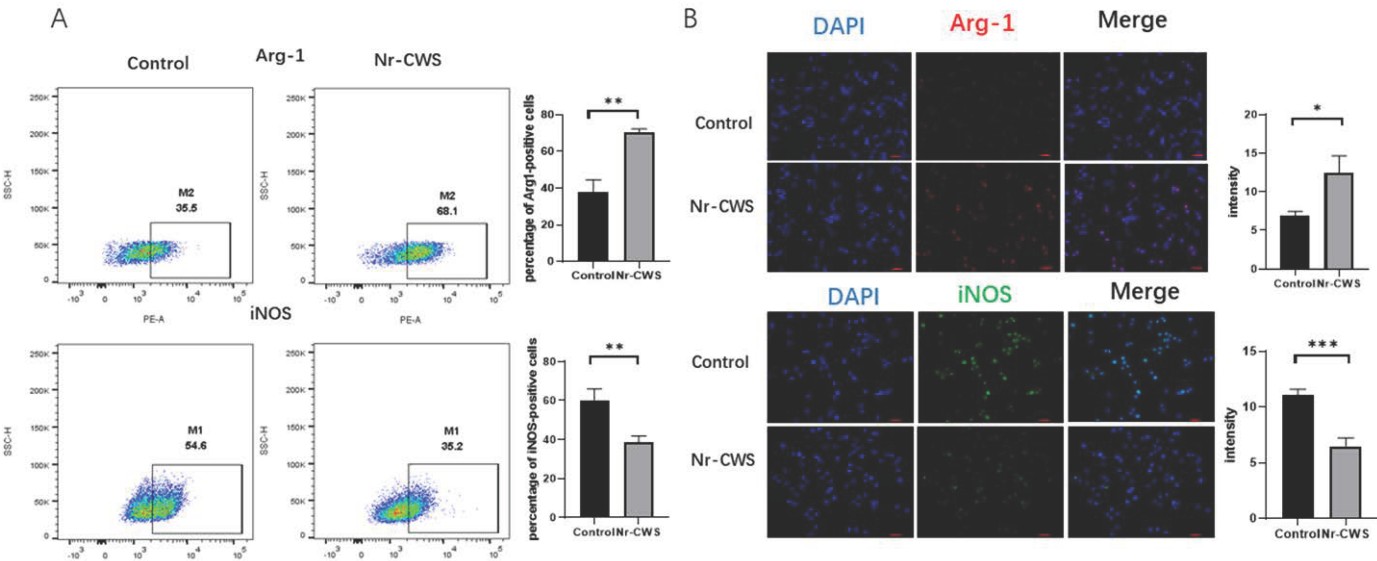

**Figure 2.** Nr-CWS can promote macrophage polarization toward the M2 phenotype. (**A**) Flow cytometry showed that the percentage of iNOS-positive cells decreased, while the percentage of Arg-1 positive cells increased upon N-CWS treatment. (**B**) The immunofluorescence results also showed that the number of Arg-1 positive cells increased dramatically while the number of iNOS-positive cells significantly decreased after Nr-CWS treatment. (* $p < 0.05$, ** $p < 0.01$, *** $p < 0.001$ vs. control, paired *t* test).

Subsequently, we focused on the expression of M2typerelated cytokines. First, using ELISA, we detected significantly increased IL10 and TGF-β1 secretion in the Nr-CWS group compared to the DMEM group (Figure 3A). Then, we analyzed the gene expression of the related proteins by qPCR. The results showed that the gene expression of the M2-type-related proteins IL10, TGF-β1, and Arg-1 were significantly increased in the Nr-CWS group. There was no significant difference in the iNOS (Figure 3B). Finally, we used Western blotting to investigate the PI3K-Akt-MTOR pathway, which can promote the M2 polarization of macrophages [12–14]. We observed that the phosphorylated PI3K/Akt/mTOR level significantly increased following the application of the Nr-CWS. Therefore, we hypothesized that the Nr-CWS could affect the polarization of macrophages through the PI3K/AKT/mTOR pathway. Furthermore, we determined the protein expression level of TGF-β1, and the results were consistent with the ELISA results. The Nr-CWS group significantly promoted the expression of TGF-β1 in macrophages (Figure 3C).

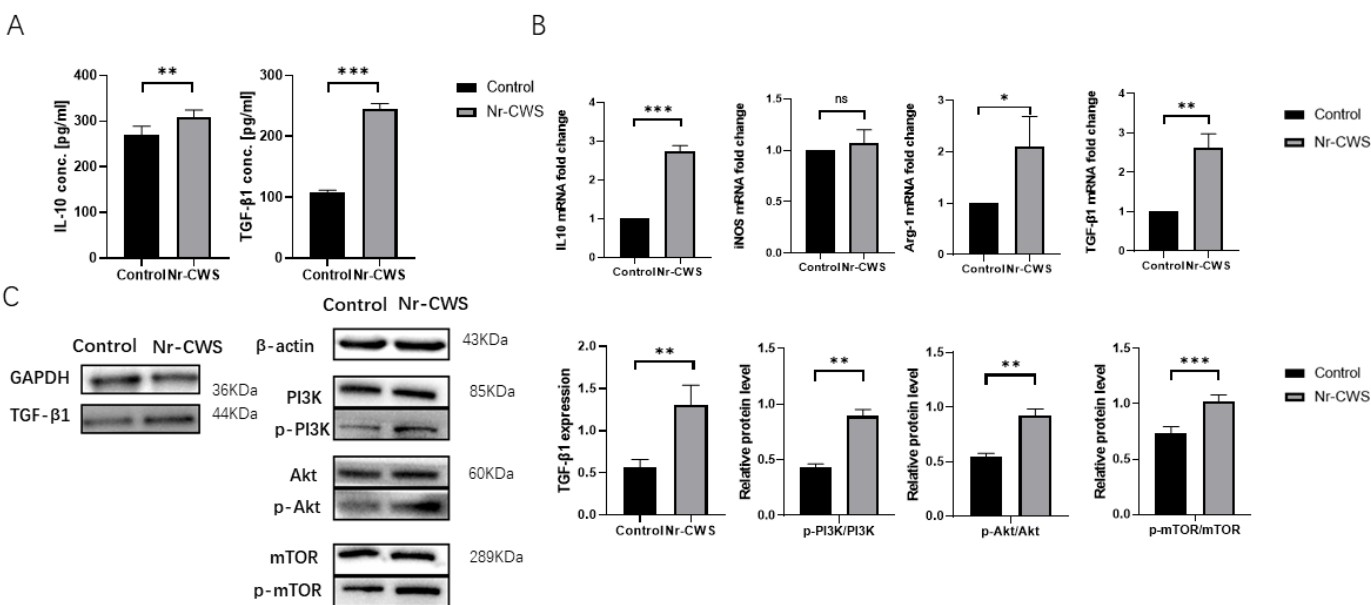

**Figure 3.** Expression of genes and proteins secreted by M2 macrophages. (**A**) An ELISA showed increased IL10 and TGF-β1 secretion in the Nr-CWS group. (**B**) qPCR showed increased expression of the M2 macrophage-related genes ARG-1, IL10, and TGF-β1. (**C**) Protein levels of PI3K, Akt, mTOR, phosphorylated PI3K, Akt, mTOR and TGF-β were analyzed by western blotting. β-actin and GAPDH were used as internal references. The statistics were analyzed in grayscale with ImageJ (ns—not significant, * $p < 0.05$, ** $p < 0.01$, *** $p < 0.001$ vs. control, paired *t* test).

### *3.2. In Vivo Experiments*

#### 3.2.1. Nr-CWS Can Enhance the Recruitment of Macrophages in the Wound and Polarize the Macrophages

In the animal experiments, we removed and fixed the tissue samples from the mice on days three, seven, and ten. Then, we dehydrated and embedded sections for relevant histopathological examination. When observing the H&E-stained sections on day three, we found that the Nr-CWS group had more inflammatory cell infiltration than the control group. (Figure 4A) We then analyzed the number and polarization of the macrophages through immunofluorescence, using CD68 as a total macrophage marker and CD163 as a specific marker for M2 macrophages. As observed in the immunofluorescence, the number of macrophages in the Nr-CWS group was significantly higher than in the control group on day three. We calculated the CD163/CD68 to derive the proportion of M2 macrophages and found that the proportion of M2 macrophages in the Nr-CWS group was significantly higher than in the control group on day seven (Figure 5A).

#### 3.2.2. Nr-CWS Can Promote Wound Healing

Finally, we evaluated the indices related to wound healing in mice. First, we filmed and recorded the wound changes in mice on days zero, two, four, seven, and ten. We calculated the wound area and healing rate by the formula: Healing rate = (original wound area—current measured area) ÷ original wound area × 100%. The Nr-CWS group showed a significantly higher healing rate and smaller wound area at each time point (Figure 6A). Then, we performed Masson staining on the wound tissue specimens on days three, seven, and ten to explore the collagen deposition in each group. The section dyed blue in the figure indicates the collagen fiber deposited in the wound. By calculating the collagen volume fraction, we found that the collagen volume fraction of the Nr-CWS group was significantly higher than that of the control group (Figure 4B). We also observed the H&E staining on day seven. The fibroblasts from the Nr-CWS group increased significantly compared with the control group. In addition, we calculated the thickness of the wound tissue and found that the thickness of the Nr-CWS group was significantly increased compared with the control

group (Figure 4A). These results indicate that the Nr-CWS accelerated the regeneration of wound granulation.

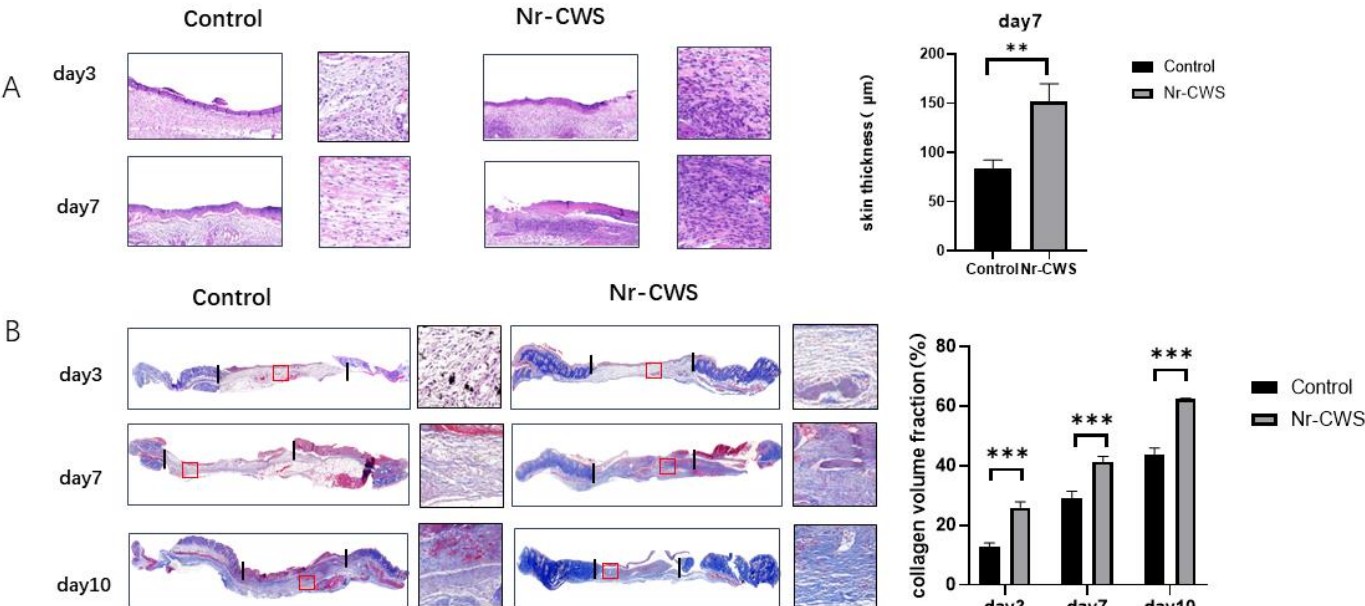

**Figure 4.** Injured tissues of mice on Days 3, 7 and 10 were removed, fixed and dehydrated, embedded in sections and subjected to relevant pathological examinations. (**A**) H&E staining of traumatic tissues on Days 3 and 7. (**B**) Masson staining of traumatic tissues of mice on Days 3, 7 and 10. (** $p < 0.01$, *** $p < 0.001$ vs. control, two-way ANOVA).

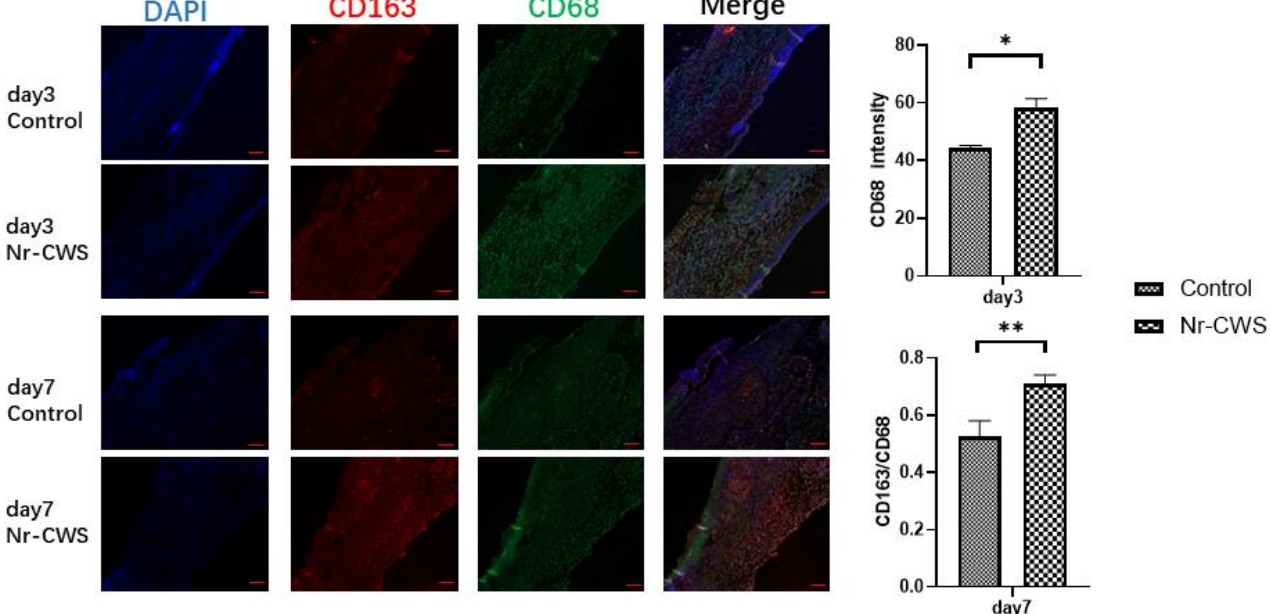

**Figure 5.** Injured tissues of mice on Days 3, 7 and 10 were removed, fixed and dehydrated, embedded in sections and subjected to relevant pathological examinations. Immunofluorescence staining of injured tissues with CD68-labeled macrophages and CD163-labeled M2 macrophages on Days 3 and 7 (* $p < 0.05$, ** $p < 0.01$ vs. control, paired $t$ test).

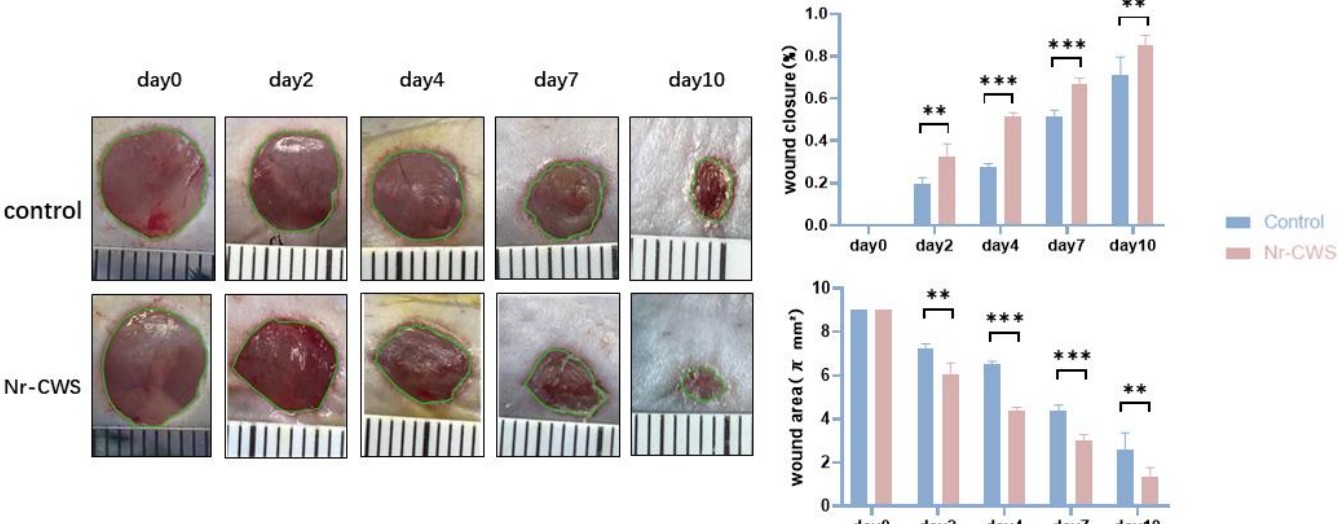

**Figure 6.** Wounds were photographed at the designated times, and Day 0 photographs were taken immediately after the injury. The healing rate was calculated as follows: Healing rate = (original wound area − current measured area) ÷ original wound area × 100% (** $p < 0.01$, *** $p < 0.001$ vs. control, two-way ANOVA).

## 4. Discussion

The Nr-CWS is an immunomodulator with therapeutic potential and has been used in antitumor-related diseases [15]. Notably, some researchers have reported that Nr-CWSs can regulate the physiological properties related to macrophages [16]. Macrophages are mononuclear phagocytes established during embryonic development. They are derived from the yolk sac or fetal liver. They are also recruited from the blood and bone marrow under inflammatory conditions (e.g., tissue repair in wound healing). Macrophages display critical regulatory activities at all stages of wound repair, where they can secrete relevant chemokines, matrix metalloproteinases, and other inflammatory mediators that drive the initial postinjury cellular response [17]. Most importantly, macrophages can transition from being inflammatory to anti-inflammatory through a phenotypic switch [18]. Therefore, macrophages represent a potentially important therapeutic target for wound healing.

Related studies have shown that reducing inflammation may be detrimental to the regeneration of wound tissue. The nonselective elimination of macrophage populations or affecting molecules for their recruitment delays wound healing [19–21]. Early recruitment of macrophages is essential for inflammation and for the establishment of the healing cascade [22]. Therefore, instead of a broad-stroke anti-inflammatory approach, the best process may be following the action of the macrophages and thus allowing tissue recovery [23,24]. Here, through an in vivo study using H&E staining and immunofluorescence, we found that Nr-CWSs can promote early macrophage recruitment in mouse wounds. The CCK8 and transwell assays demonstrated that the Nr-CWS could promote macrophage proliferation and migration in vitro. After macrophages are recruited to the wound surface, they function as scavenger cells to phagocytose cell debris, invading organisms, neutrophils, and apoptotic cells. In addition, we found that the Nr-CWS can promote macrophage phagocytosis by flow cytometry. These results illustrate that Nr-CWSs can promote the physiological activity of wound macrophages, which lays a foundation for the subsequent healing phase to proceed faster.

Macrophages exhibit significant phenotypic diversity, and microenvironmental factors influence their secretory profile and functional properties. The two major macrophage populations, classically activated (M1) and alternatively activated (M2) macrophages. M1 produce pro-inflammatory mediators, while M2 exhibit immunomodulatory properties. In the early phase, macrophages exhibit the M1 phenotype and release pro-inflammatory factors such as TNF-a and IL-1β. However, if the M1 stage persists, it may lead to tissue

damage. Therefore, M2 macrophages secrete large amounts of growth factors such as IL-10 and TGF-β, which contribute to tissue repair, remodeling, angiogenesis, and maintenance of homeostasis in vivo [25–27].

To investigate the effect of Nr-CWSs on macrophage polarization, we performed labeling assays on mouse tissues and RAW264.7 macrophages. In this experiment, Arg-1 and CD163 were used as markers of M2 macrophages, while iNOS was used as an M1 marker [24]. We analyzed the proportion of macrophages and found that the Nr-CWS significantly promoted polarization towards M2 compared with the control group. In addition, we found that the gene and protein expression of TGF-β1, IL10, and Arg-1 were elevated. The PI3K/AKT/mTOR pathway phosphorylated protein levels were increased too. We hypothesize that the Nr-CWS may prompt macrophages to polarize through this signaling pathway. Moreover, we found that the wounds of mice in the Nr-CWS group could promote the deposition of collagen fibers and the formation of granulation tissue. The Nr-CWS accelerated the re-epithelialization process compared with the control group, and the final skin tissue formed was closer to normal skin. We thought that the Nr-CWS might increase the secretion of relevant healing factors by promoting the polarization of M2 macrophages, thus accelerating the healing process of the wound.

There are limitations to this study, such as the small number of samples; the exploration of the pathway did not form a closed-loop validation. Additionally, the mechanism of the Nr-CWS on primary skin cells, such as fibroblasts, endothelial cells, and human keratinocytes, is worth exploring.

However, immunomodulators such as Nr-CWSs provide new therapeutic ideas, such as promoting wound healing by upregulating immune mechanism responses. Some studies have suggested that diabetic wounds exhibit a significant delay in macrophage infiltration early after injury [27,28]. They used a chemoattractant, CCL2, significantly promoting diabetic wound healing by restoring the macrophage response. Therefore, the restoration of proper kinetics of the macrophage response may be able to initiate the subsequent healing phase rapidly. We hope that Nr-CWSs could play a similar immune-driven role as CCL2, and it is worth further study.

In summary, we found that the Nr-CWS, as an immunomodulator, can increase the proliferation, migration, and phagocytosis of macrophages, and the recruitment of macrophages at the wound surface. Then, it can polarize macrophages toward M2 and increase the expression of pro healing factors such as IL10 and TGF-β1, thus accelerating the healing of skin wounds.

**Author Contributions:** Conceptualization, G.L. and K.H.; methodology, Y.X.; software, Y.X.; validation, P.D. and Y.L.; formal analysis, X.L.; investigation, K.H.; resources, G.L.; data curation, X.L.; writing—original draft preparation, K.H. and Y.X.; writing—review and editing, K.H. and Y.X.; visualization, P.D. and Y.L.; project administration, G.L.; funding acquisition, G.L. All authors have read and agreed to the published version of the manuscript.

**Funding:** This study was supported by the Clinical Frontier Technology Project of Jiangsu Provincial Key Research and Development Program (Social Development) (BE2018626).

**Institutional Review Board Statement:** The animal study protocol was approved by Ethics Committee of Jiangnan University (protocol code: JN.No20210415c1350920[086]; date of approval: 24 March 2021) for studies involving animals.

**Data Availability Statement:** Not applicable.

**Conflicts of Interest:** The authors declare that there are no conflict of interest.

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
