# Peer review of "The Nocardia Rubra Cell Wall Skeleton Regulates Macrophages and Promotes Wound Healing"

_cimb, doi:10.3390/cimb44120408_

Round 1

Reviewer 1 Report

Kai Hu et al’s manuscript “The Nocardia Rubra Cell Wall Skeleton Regulates Macro-2 phages and Promotes Wound Healing” tested the effect of Nocardia rubra cell wall skeleton (Nr-CWS) in an in vitro model and an in vivo model, trying to explore the function of Nr-CWS in macrophage reprogramming and wound healing. However, there are some major issues to be addressed.

1.     At the beginning of the introduction, the authors mentioned that the Nr-CWS showed anti-tumor effect. The reference 1 showed that it increased the cytokines IL-6, IL-12, TNF-É‘, and IL-1β secreted by dendritic cells and macrophages, inhibit the proliferation, which was totally opposite to this manuscript. Why is that?

2.     Figures were not well organized. The labels did not match the text, which makes the reading and interpretation very confusing. These mistakes were not acceptable. The authors should carefully check it and correct it accordingly. For examples, in lines 222-225, the authors mentioned that they quantitated the ratio of CD163/CD68 in figure 4C, however, there was no Figure 4C. In fig4B, the figures all overlapped and not well organized.

3.     Fig1C, the FITC-dextran staining did not show shift, which did not support their conclusion that the FITC-positive cells was significantly higher in the Nr-CWS group.

4.     Fig2A, what was the gating strategy? The authors should show their gating strategy by adding FMO controls. Fig2B, Arg1 panel, why were the Arg1+ cells not positive for DAPI?

5.     In figure 4A, the TGFbeta level was total or active form?

6.     Fig 3C, the WB of GAPDH and TGFbeta did not seem like the same ones as the raw pictures. The authors should provide the full WB images with the ladders as the supplemental data.

7.     Fig 5C, the intensity of DAPI was very dim, and the background of CD163 was really high. The signal did not seem like specific staining. Did the authors use the same setting and intensity for all the pictures?

8. Fig 6, the would area of day should be quantitated and showed in the bar graph too.

Author Response

Dear editor

On behalf of my co-authors, we thank you very much for allowing us to revise our manuscript. We appreciate your time and effort on this manuscript. You read it carefully and gave us many constructive suggestions to improve the quality of our manuscript. I have carefully read your comments and revised them. The following is my reply. I hope you can spare some time to guide me.

1.“At the beginning of the introduction, the authors mentioned that the Nr-CWS showed anti-tumor effect. The reference 1 showed that it increased the cytokines IL-6, IL-12, TNF-É‘, and IL-1β secreted by dendritic cells and macrophages, inhibit the proliferation, which was totally opposite to this manuscript. Why is that?”

Reply: Thank you for your review. Reference 1 showed that Nr-CWS could promote the activity of dendritic cells and macrophages and reduce their apoptosis, coinciding with our results. The inhibited cells were HeLa and SiHa cervical carcinoma cell lines. Nr-CWS promotes IL6,IL-12, TNF-É‘ and IL-1β secreted by macrophages. These cytokines may inhibit the HeLa and SiHa cell lines' growth.

Some other studies also indicated that Nr-CWS could promote the proliferation and phagocytic activity of macrophages.

Reference:

1.Mine Y, Watanabe Y, Tawara S, Nonoyama S, Yokota Y, Kikuchi H. In vivo activation of functional properties in mouse peritoneal macrophages by Nocardia rubra cell wall skeleton. Arzneimittelforschung. 1986;36(11):1651-1655.

2.Wang Y, Hu Y, Ma B, et al. Nocardia rubra cell wall skeleton accelerates cutaneous wound healing by enhancing macrophage activation and angiogenesis. J Int Med Res. 2018;46(6):2398-2409. doi:10.1177/0300060518764210)

2. “Figures were not well organized. The labels did not match the text, which makes the reading and interpretation very confusing. These mistakes were not acceptable. The authors should carefully check it and correct it accordingly. For examples, in lines 222-225, the authors mentioned that they quantitated the ratio of CD163/CD68 in figure 4C, however, there was no Figure 4C. In fig4B, the figures all overlapped and not well organized.”

Reply:Thank you for reminding me, I have rearranged the pictures and articles.

3” Fig1C, the FITC-dextran staining did not show shift, which did not support their conclusion that the FITC-positive cells was significantly higher in the Nr-CWS group.”

Reply:I'm sorry for not presenting the data perfectly. I have remade the pictures,and set up the gating strategy.

4” Fig2A, what was the gating strategy? The authors should show their gating strategy by adding FMO controls. Fig2B, Arg1 panel, why were the Arg1+ cells not positive for DAPI?”

Reply:I am sorry for some controversial pictures. I have recalculated the flow cytometry data and made the graphs;the gating strategy also has been added.

The DAPI looks negative probably due to the low number of cells in the view, I have changed another parallel data for this group.

5.” In figure 4A, the TGF-beta level was total or active form?”

Reply: It was total form.

6.Fig 3C, the WB of GAPDH and TGF-beta did not seem like the same ones as the raw pictures. The authors should provide the full WB images with the ladders as the supplemental data.

Reply: The image is partially taken from the original photo but does not look the same because the image size was changed during processing. Here are the full WB images with the ladders.

7.”Fig.5C, the intensity of DAPI was very dim, and the background of CD163 was really high. The signal did not seem like specific staining. Did the authors use the same setting and intensity for all the pictures?”

Reply: Considering that the background of CD163 may be high due to some non-specific staining, we re-selected the parallel immunofluorescence data on day7 and performed the statistical calculation. The results are shown in the figure below. All the settings and intensity of the pictures are the same.

8.” Fig 6, the would area of day should be quantitated and showed in the bar graph too.”

Reply: I’ve revised the images, and the wound area of the day has been shown in the figure.

Please allow me once again to express my sincerest greetings to you and thank you for your time and effort in our article!

Yours sincerely

Guozhong Lv

Reviewer 2 Report

In this study, the authors investigate the effect of  Nocardia rubra cell wall skeleton (Nr-CWS) on cutaneous wound healing and in vitro macrophage functions.  They found the Nr-CWS treatment accelerated wound healing probably by enhancing the M2 polarization of wound macrophages.

The current manuscript seems to be the continuation of a previous paper but the reference for that is missing: Yi Wang, J Int Med Res. 2018 Jun; 46(6): 2398–2409. Here, the authors performed similarly in vivo wound-healing experiments and found that Nr-CWS treatment accelerates the process. 

Comments:

Why dextran was used in the phagocytosis assay? Apoptotic or necrotic cells would mimic the in vivo conditions better.

Line 118 "The effect of FITC dextran on cell phagocytosis was analyzed by flow cytometry" No, the effect of Nr-CWS on the uptake of FITC dextran was analyzed. 

Line 107 "Nr-CWS was dissolved in 2 ml saline and then added to the well plates at 0%, 5%, 10%, 25% and 50% concentrations..." How much Nr-CWS was dissolved?  The weight is missing.

Figure 1, panel A: the axis title should be cell number (% of control). Viability cannot be 150%.

In panel C, the representative histogram is not showing that only 15% of the control RAW cells were phagocytosing. 

In line 202, the authors start to describe the in vivo experiments but 10 lines below they turn back to the cell line experiments. This paragraph could go after the in vitro experiment section.

The order of the figures is random. Figure 4 comes after figure 1.

Figure 3 panel B, the Y-axis titles should be positioned correctly.

Figure 5, panels A and B are missing.

Which company is Therm o, mentioned in the Materials and methods section?

The font size should be increased in the figures. The text is blurry too.

Typographical errors should be corrected:

E.g. Line 113  "Raw264.7 cells (1×104 cells/well) were plated on 96-well plates..."

The English style should be improved. The text contains long sentences that are hard to understand. For e.g. "M1 macrophages are classically activated and usually accumulate at the wound surface during the early inflammatory phase of wound healing, and M2 macrophages are alternatively activated and usually accumulate at the wound site during the repair phase of wound healing, a phenotypic transformation that is thought to be necessary for wound healing."

Author Response

Dear editor

On behalf of my co-authors, we thank you very much for allowing us to revise our manuscript. We appreciate your time and effort on this manuscript. You read it carefully and gave us many constructive suggestions to improve the quality of our manuscript. I have carefully read your comments and revised them. The following is my reply. I hope you can spare some time to guide me.

1.“Why dextran was used in the phagocytosis assay? Apoptotic or necrotic cells would mimic the in vivo conditions better.”

Reply: This experimental method is based on literature reading. We have reviewed many experiments on the phagocytosis activity of macrophages and found that this experiment is highly feasible, and the results are precise. Here are some references. We remade the graphs and set the gating strategy to display the results more intuitively.

Reference:

1.Yu Q, Nie SP, Li WJ, et al. Macrophage immunomodulatory activity of a purified polysaccharide isolated from Ganoderma atrum. Phytother Res. 2013;27(2):186-191. doi:10.1002/ptr.4698

2.Sharma L, Wu W, Dholakiya SL, et al. Assessment of phagocytic activity of cultured macrophages using fluorescence microscopy and flowcytometry.MethodsMolBiol.2014;1172:137-145.doi:10.1007/978-1-4939-0928-5_12

3.Yi Z, Guo S, Hu X, et al. Functional modulation on macrophage by low dose naltrexone (LDN). Int Immunopharmacol. 2016;39:397-402. doi:10.1016/j.intimp.2016.08.015

2. “Line 107 "Nr-CWS was dissolved in 2 ml saline and then added to the well plates at 0%, 5%, 10%, 25% and 50% concentrations..." How much Nr-CWS was dissolved? The weight is missing.”

Reply:According to the instructions, 60ug of Nr-CWS was dissolved in 2ml of normal saline.

3”Figure 1, panel A: the axis title should be cell number (% of control). Viability cannot be 150%.”

Reply:Thanks for your reminding. I have revised the chart.

4“In panel C, the representative histogram is not showing that only 15% of the control RAW cells were phagocytosing“.

Reply: I have made the new flow cytometry figures. Please kindly check!

5.”In line 202, the authors start to describe the in vivo experiments but 10 lines below they turn back to the cell line experiments. This paragraph could go after the in vitro experiment section.”

Reply: I have put the results of the in vivo experiment behind the cell experiment. Here is the revised title. The specific content is in the text.

In vitro experiments

3.1Nr-CWS enhances the proliferation, migration, and phagocytosis of RAW264.7 macrophages

3.2. Nr-CWS can polarize macrophages toward the M2 type and promote the expression of related cytokines.

In vivo experiments

3.3 Nr-CWS can enhance the the recruitment of macrophages in the wound and polarize the macrophages

3.4 Nr-CWS can promote wound healing

6.“The order of the figures is random.

Figure 4 comes after figure 1.

Figure 3 panel B, the Y-axis titles should be positioned correctly.

Figure 5, panels A and B are missing”.

Reply: All the above mistakes have been corrected. I'm sorry for my negligence.

7.“Which company is Therm o, mentioned in the Materials and methods section?”

Reply: It’s “Thermo”, I'm sorry for the spelling mistake.

8.”The font size should be increased in the figures. The text is blurry too.”

Reply: I have enlarged the font and rearranged it to make it more legible.

9.Typographical errors should be corrected: E.g. Line 113 "Raw264.7 cells (1×104 cells/well) were plated on 96-well plates...”

Reply: I have revised the typographical error.

10.“The English style should be improved. The text contains long sentences that are hard to understand. For e.g. "M1 macrophages are classically activated and usually accumulate at the wound surface during the early inflammatory phase of wound healing, and M2 macrophages are alternatively activated and usually accumulate at the wound site during the repair phase of wound healing, a phenotypic transformation that is thought to be necessary for wound healing."

Reply: I have revised part of my English style, and some complex sentences have been simplified.May I ask that we can use the polishment recommended by your paper if the article is accepted? I wish the paper would be better!Thank you!

Please allow me once again to express my sincerest greetings to you and thank you for your time and effort in our article!

Yours sincerely

Guozhong Lv

Round 2

Reviewer 1 Report

The authors somehow improved their manuscripts (i.e. reorganizing their figures and including some new pictures). However, some major questions are still yet to be addressed. 

1.     In ref 1, the Nr-CWS skewed the macrophages towards an M1-like phenotype (increased cytokines IL-6, IL-12, TNF-É‘, and IL-1β) and inhibited the proliferation of the macrophages and dendritic cells, while in this manuscript, the compound showed totally opposite effect by reprogramming the macrophages to an M2-like phenotype and increasing its proliferation. The authors did not address this question.

2.     The authors changed the illustration of the flow cytometry graphs of Fig 1C and 2A. However, it is very confusing that why the percentages were also changed significantly. The histogram graphs of their version 1 did not match with the do plots in this new version.

3.     The authors mentioned that the full WB image with ladder was provided, however, I was not able to find them.

Author Response

Dear reviewers:

Thank you again for taking the time and effort to review our paper. We appreciate your carefulness and conscientiousness. Your suggestions are precious and helpful for revising and improving our paper. The following is my response to your new comments.

1.In ref 1, the Nr-CWS skewed the macrophages towards an M1-like phenotype (increased cytokines IL-6, IL-12, TNF-É‘, and IL-1β) and inhibited the proliferation of the macrophages and dendritic cells, while in this manuscript, the compound showed totally opposite effect by reprogramming the macrophages to an M2-like phenotype and increasing its proliferation. The authors did not address this question.

Reply:Actually,we were wondering the same thing during the research. After some literature queries,we summarize the following three possible reasons.

(1)  The phenotypes of macrophages are diverse

Classic theories generally agree that macrophages are divided into two phenotypes, M1 and M2. However, some studies [1] suggested that a more informative foundation for macrophage classification should be based on the fundamental macrophage functions involved in maintaining homeostasis. They proposed three functions: host defense, wound healing, and immune regulation; classifying macrophages according to these functions provides three essential macrophage populations analogous to the three primary colors in a color wheel. This spectrum means that macrophages may have multiple functions. There may be more phenotypes in macrophages

For example,M2b macrophages, known as regulatory macrophages, can express pro-inflammatory cytokines (IL-1β, IL-6, TNF-α).M2b cells also express and secrete substantial amounts of the anti-inflammatory cytokine IL-10 and low levels of IL-12, which is the functional converse of M1 cells.[2]

In addition, the amount of secretion factor is also a consideration; If pro-inflammatory factors are on the rise but the total amount is not as large as pro-healing factors, the proportion of M1 in the cell population may be less than M2.

(2)  Polarization is a dynamic process

Macrophage polarization is a continuum of functional states; once a macrophage adopts a phenotype, it still retains the ability to change in response to new environmental influences. This result means that polarization is not static but a dynamic process,the ratio of M1 and M2 will change, too. However, our vitro experiments only studied the macrophage status after 24h of Nr-CWS treatment. In the vivo experiments,the replacement time of Nr-CWS is also fixed. We observed a fixed point in time rather than a dynamic change of macrophages. So there may be incomplete results.[3]

(3)  Different source of macrophages lead to changes in the direction of polarization

In ref1, the cells researchers used were macrophages isolated from human peripheral blood lymphocytes. However, our experiments were based on mouse macrophages. Macrophages are susceptible to the microenvironment, and the difference in source and culture environment may affect their differentiation. Besides,In different diseases, different cell metabolism results in differences of their polarization. So we think that might be one of the reasons.[4]

Reference:

  1. Mosser DM, Edwards JP. Exploring the full spectrum of macrophage activation [published correction appears in Nat Rev Immunol.2010 Jun;10(6):460]. Nat Rev Immunol. 2008;8(12):958-969. doi:10.1038/nri2448
  2. Wang LX, Zhang SX, Wu HJ, Rong XL, Guo J. M2b macrophage polarization and its roles in diseases. J Leukoc Biol. 2019;106(2):345-358. doi:10.1002/JLB.3RU1018-378RR
  3. Funes SC, Rios M, Escobar-Vera J, Kalergis AM. Implications of macrophage polarization in autoimmunity. Immunology. 2018;154(2):186-195. doi:10.1111/imm.12910
  4. Sun JX, Xu XH, Jin L. Effects of Metabolism on Macrophage Polarization Under Different Disease Backgrounds. Front Immunol. 2022;13:880286. Published 2022 Jul 14. doi:10.3389/fimmu.2022.880286

2.The authors changed the illustration of the flow cytometry graphs of Fig 1C and 2A. However, it is very confusing that why the percentages were also changed significantly. The histogram graphs of their version 1 did not match with the do plots in this new version.

Reply: Sorry for the misunderstanding. We slightly adjusted the cell population and gating strategy when reprocessing the original data. Therefore, the new figure is not the same as the original data, but the trend and difference between the two groups are the same as the original.

3.The authors mentioned that the full WB image with ladder was provided, however, I was not able to find them.

Reply: The receipt we sent to you in the submission system can only upload pdf/word files, so we are unsure whether you can see the materials we uploaded in the background. We have uploaded all the original WB images and makers to the system. However, it is a pity that the maker of p-mtor and p-pi3k forgot to shoot due to the operator's negligence. (They were on the same membrane.) We, therefore, provided the adnexa of β-actin and p-akt on the same membrane for examination. Their protein masses were (β-actin 43KDa; P - akt 60KDa; P - pi3k 85 KDa; p-mtor 289 KDa). Thank you!

It is my honor to receive your advice again! Please allow me to express my sincere thanks and best wishes to you!

Reviewer 2 Report

The quality of the revised manuscript has been significantly improved, yet the English language still needs to be edited.

E.g. instead of "Performed the images under a Leica inverted phase contrast microscope. " you may write "Photos were taken using an inverted Leica phase contrast microscope.".

The sentence "Next, we used ELISA, qPCR, and Western blotting to examine the protein and gene expression related to factors." makes no sense.

The sentence "First, we used ELISA to detect the expression of IL10 and TGF-β1, and the results showed that the secretion factors in the Nr-CWS group were significantly increased compared with those in the DMEM group." makes no sense.  Perhaps the authors meant this: "First, using ELISA, we detected significantly increased IL10 and TGF-β1 secretion in the Nr-CWS-treated group compared to the control cells.".

"The effect of FITC dextran on cell phagocytosis was analyzed by flow  cytometry..." 

As I pointed out earlier, not the effect of dextran was investigated but the impact of Nr-CWS on the phagocytosis of dextran particles.

Scientific comments:

The amount of dissolved Nr-CWS (60 micrograms) should be indicated in the text.

In Figure 2 legend and the corresponding main text, it would be more appropriate to write that the percentage of iNOS-positive cells decreased, while the percentage of Arg-1 positive cells increased upon Nr-CWS treatment. 

In line 248, what are the Nr-CWS fibrocytes? How the number of fibrocytes (or fibroblasts?) was determined?

A significant reference, describing the impact of Nr-CWS on cutaneous wound healing and macrophage activation, is still missing: Yi Wang, J Int Med Res. 2018 Jun; 46(6): 2398–2409. 

Author Response

1.The quality of the revised manuscript has been significantly improved, yet the English language still needs to be edited.

E.g. instead of "Performed the images under a Leica inverted phase contrast microscope. " you may write "Photos were taken using an inverted Leica phase contrast microscope."

The sentence "Next, we used ELISA, qPCR, and Western blotting to examine the protein and gene expression related to factors." makes no sense.

The sentence "First, we used ELISA to detect the expression of IL10 and TGF-β1, and the results showed that the secretion factors in the Nr-CWS group were significantly increased compared with those in the DMEM group." makes no sense.  Perhaps the authors meant this: "First, using ELISA, we detected significantly increased IL10 and TGF-β1 secretion in the Nr-CWS-treated group compared to the control cells."

"The effect of FITC dextran on cell phagocytosis was analyzed by flow cytometry..." 

As I pointed out earlier, not the effect of dextran was investigated but the impact of Nr-CWS on the phagocytosis of dextran particles.

Reply: Thank you very much! I have revised all the comments you listed and reviewed the whole text again.

Scientific comments:

2.The amount of dissolved Nr-CWS (60 micrograms) should be indicated in the text.

Reply: Thank you for reminding me. I have indicated in the methods.

“Nr-CWS (60 micrograms) was dissolved in 2 ml saline. All subsequent experiments were the same. “[line75]

3.In Figure 2 legend and the corresponding main text, it would be more appropriate to write that the percentage of iNOS-positive cells decreased, while the percentage of Arg-1 positive cells increased upon Nr-CWS treatment.

Reply: Thank you for your advice! I've revised it both in the text and in the illustrations.

“Analysis by flow cytometry showed that the percentage of iNOS-positive cells decreased, while the percentage of Arg-1 positive cells increased upon Nr-CWS treatment. (Fig. 2A) The immunofluorescence results also showed that the number of Arg-1-positive cells increased dramatically while the number of iNOS-positive cells significantly decreased in the Nr-CWS group. (Fig.2B)”

“Fig. 2. Nr-CWS can promote macrophage polarization toward the M2 phenotype (A). Flow cytometry showed that the percentage of iNOS-positive cells decreased, while the percentage of Arg-1-positive cells increased upon Nr-CWS treatment. (B). The immunofluorescence results also showed that the number of Arg-1-positive cells increased dramatically while the number of iNOS-positive cells significantly decreased after Nr-CWS treatment. (*P < 0.05, **P < 0.01, ***P < 0.001 vs. control, paired t test).”

4.In line 248, what are the Nr-CWS fibrocytes? How the number of fibrocytes (or fibroblasts?) was determined?

Reply: Sorry for not describing it clearly. It is the fibroblasts from the Nr-CWS group. We invited the doctor director of the pathology department to perform a diagnostic evaluation of H&E sections. It is generally believed that the fusiform cells in the wound are fibroblasts. Based on their clinical experience, the degree of fibroblast proliferation was classified as mild, moderate, or severe. In our experiment, the pathology doctor judged that the degree of fibroblast growth was higher in the Nr-CWS group than in the control group.

5. A significant reference, describing the impact of Nr-CWS on cutaneous wound healing and macrophage activation, is still missing: Yi Wang, J Int Med Res. 2018 Jun; 46(6): 2398–2409.

Reply: I have added the literature to the references, thank you for your careful reading.

It is my honor to receive your advice again! Please allow me to express my sincere thanks and best wishes to you!
